# Surviving to Acute Myocardial Infarction: The Role of Psychological Factors and Alexithymia in Delayed Time to Searching Care: A Systematic Review

**DOI:** 10.3390/jcm10173813

**Published:** 2021-08-25

**Authors:** Federica Sancassiani, Roberta Montisci, Antonio Preti, Pasquale Paribello, Luigi Meloni, Ferdinando Romano, Antonio E. Nardi, Mauro Giovanni Carta

**Affiliations:** 1Department of Medical Sciences and Public Health, University of Cagliari, Asse Didattico E. SS 554 Bivio Sestu, 09042 Monserrato, CA, Italy; federicasancassiani@yahoo.it (F.S.); pasqualeparibello@gmail.com (P.P.); meloni.unica@gmail.com (L.M.); maurogcarta@gmail.com (M.G.C.); 2Department of Neurosciences, University of Turin, 10126 Turin, Italy; antonio.preti@unito.it; 3Department of Public Health and Infectious Diseases, University of Rome “La Sapienza”, 00185 Rome, Italy; ferdinando.romano@uniroma1.it; 4Laboratory of Panic and Respiration, Institute of Psychiatry of Federal University of Rio de Janeiro (IPUB/UFRJ), Rio de Janeiro 22290-140, Brazil; antonioenardi@gmail.com

**Keywords:** acute myocardial infarction, alexithymia, care-seeking behavior, pre-hospital delay, decisional delay, psychological factors

## Abstract

The time from symptom onset to reperfusion is a critical determinant of myocardial salvage and clinical outcomes in patients with acute myocardial infarction (AMI). This time period could be delayed if people do not seek help promptly and/or if the health system is not efficient in responding quickly and attending to these individuals. The aim of this study was to identify psychological factors associated with pre-hospital delay (PHD) or patients’ decisional delay (PDD) in people with an ongoing AMI. A search in PubMed/Medline from 1990 to 2021 with the keywords “pre-hospital delay” OR “prehospital delay” OR “patient delay” OR “decisional delay” OR “care seeking behavior” AND “psychological factors” OR “alexithymia” AND “myocardial infarction” was performed. Thirty-six studies were included, involving 10.389 patients. Wrong appraisal, interpretation and causal beliefs about symptoms, denial of the severity of the symptoms and high levels of alexithymia were found related to longer PHD or PDD. Alexithymia may be an overarching construct that explains the disparate findings of the studies exploring the role of psychological factors in PHD or PDD. Further studies are needed in order to analyse the role of alexithymia in patients with risk factors for AMI to prevent delay.

## 1. Introduction

In healthcare, “time to treatment” is an increasingly important dimension in order to prevent complications and to promote a successful management of many illnesses, such as acute myocardial infarction (AMI). AMI is a medical emergency for which its treatment is time-dependent, especially when it occurs with ST segment-elevation (STEMI). In this case, when recanalization of the culprit coronary artery occurs within the first hours of symptoms’ onset, reperfusion is critical for myocardial salvage and survival [1].

Delay in seeking care is still an important issue in patients with an ongoing AMI, as it limits the timely access to coronary reperfusion. Patients should enter quickly into the healthcare system to receive the most benefit from reperfusion treatment. However, the largest proportion (60–80%) of the total ischemic time (from AMI symptoms onset to the recanalization of the culprit coronary artery) occurs before hospitalization, and it is determined by patients’ referral in seeking medical help (patients’ decisional delay—PDD), along with the time needed in the emergency medical system to treat the patient from the first intervention by a medicalized ambulance to the hospital door arrival (pre-hospital delay—PHD) [2].

Patients’ delay in seeking help, as well as further delay in investigations and interventions, depends on various factors and becomes particularly relevant in the setting of AMI. Indeed, early interventions can save lives and reduce long-term morbidity [3,4].

Over the past 30 years, many studies have tried to identify the factors that influence patients’ decisional time in healthcare seeking during an ongoing AMI. Most of these studies looked at socio-demographic and clinical variables. Factors such as older age, female sex, lower socio-economic status, previous infarction or other heart diseases, atypical symptoms presentation, comorbidities, symptoms onset context (i.e., being/living alone) and first consultation with a family member or a general physician appear to increase patients’ decisional time, but they are not sufficient for explaining delay [2].

Psychological factors were found to contribute to explain delay independently or in interaction with other variables [5,6,7,8,9]. Psychological factors are individual-level processes and meanings, such as beliefs, that influence mental states. Among these individual-level processes, there are also personality traits that contribute to establish and direct behavioral patterns that could be very different among different individuals. Psychological factors are also involved in the decision-making process, particularly as far as events appraisal and awareness are concerned.

The notion that these variables could affect health behaviors and vulnerability to diseases has been widely promoted in the psychosomatic field [10]. Research has extensively studied some personality patterns that do not cause diseases directly but work as intermediaries in moderating and mediating both the appraisal and patients’ self-management of symptoms [10]. In a previous review [5], recognition and treatment of psychological factors relevant to the acute pre-hospital and in-hospital phases of AMI were summarized. Several emotions and personality features, particularly type A behavior, denial and low levels of somatic and emotional awareness as characteristics of people with high levels of alexithymia, were indicated as both risk factors and consequences of acute AMI. They have been linked with excessive treatment-seeking delay for patients with an ongoing AMI, and their identification and treatment were proposed as crucial in providing effective care for AMI patients [5]. Regarding the association with patients’ delay in seeking help during an ongoing AMI, the appraisal and awareness of symptoms, as well as coping strategies, seem to play an important role [2,5,6,7,9].

Among personality traits, alexithymia is currently recognized as a risk factor for medical, psychiatric or behavioral problems [11,12,13]. Alexithymia refers to a cognitive deficit characterized by three main facets: difficulties in identifying feelings (DIF), difficulties in describing feelings (DDF) and externally oriented thinking, with an inclination to focus the details of external events (EOT). Consequently, people with high alexithymia recognize emotions mainly by their biological components, but they are poorly able to use psychological skills such as images, thoughts, imagination and words to represent and communicate emotions and their related feelings [14,15]. Some research pointed out that high alexithymia, particularly DDF facet, seems to be related to both the hypothalamic–pituitary–adrenal (HPA) axis and the sympathetic-adrenal medullary (SAM) system markers before social stress exposure, particularly with an increased basal salivary cortisol level, possibly by affecting the anticipatory cognitive appraisal of situations [16] as well as with smaller electrodermal responses [17].

Considering that an ongoing AMI is a stressful event, seeking appropriate medical care to properly manage it is another source of distress, as it requires a further exposition for the patient to ask for help from someone. It can be hypothesized that stressing situations such as these could be challenging for people with high alexithymia, as they are inclined to misinterpreting their bodily reactions—because they usually experience attenuated bodily reactions—and it makes difficult to anchor their feelings correctly [17]. In this manner, alexithymia may influence the patients’ decisional delay in seeking appropriate medical care during AMI by impairing awareness, appraisal and management of AMI. This relation, however, has been poorly investigated [5,18].

This study sets out to perform a systematic review of all research that pointed out the association between psychological factors involved in pre-hospital delay (PHD) with a focus on patients’ decisional delay (PDD) in order to answer the following questions: (a) which psychological factors are mostly involved in patients with ongoing AMI decisional delay and (b) is alexithymia an overarching factor able to cluster the main features of these psychological factors involved in patients’ decisional delay?

## 2. Materials and Methods

### 2.1. Data Sources

The PUBMED/Medline electronic database was used to search for articles following these inclusion criteria: (1) original papers written in English; (2) dating from 1990 to 2021; and (3) containing the key words “pre-hospital delay” OR “prehospital delay” OR “patient delay” OR “care seeking behavior” AND “psychological factors” OR “alexithymia” AND “myocardial infarction”. The last search was performed on March 2021. Review papers, meta-analyses, qualitative studies, editorials, commentaries, case reports and studies without a clear indication of PHD or PDD by minutes, hours or days, studies carried out on specific groups of people (i.e., women or men only; people belonging on a particular ethnicity), studies considering psychiatric disorders or those ruling out psychological factors as determinants of delay were excluded. The references section of the selected papers was used to find further studies.

### 2.2. Definition of Pre-Hospital Delay (PHD) and Patients’ Decisional Delay (PDD)

For the purposes of the present study and consistent with previous literature [19,20], PHD was defined as the time interval between the onset of signs and symptoms suggestive of AMI and the time of arrival at the hospital emergency department. This interval includes also the time needed for the emergency medical system to treat the patient, from the first intervention by a medicalized ambulance to hospital door arrival.

PDD was defined as the time interval between the onset of signs and symptoms suggestive of AMI and the patients’ decision to seek medical care.

PHD and PDD were generally reported by minutes, hours or days.

### 2.3. Data Extraction

Data extracted from the papers include the following: (1) study description (author, study year, country, design and sample size); and (2) information on PHD or PDD, as reported by the authors of the included studies by mean ± sd or median in minutes, hours or days. In case-control studies, cut-off scores to define patients were also reported as “prompt” or “delayers” regarding the optimal interval time to receive proper interventions to treat AMI and (3) psychological factors presumed to be associated with PHD or PDD delay.

### 2.4. Quality Assessment

The quality of each reviewed study was assessed by using Downs and Black’s checklist [21]. The quality score was calculated as the total number of applicable items from the checklist (minimum 23 and maximum 27) addressed in the paper under review. Studies with a quality score <50% were excluded from further consideration.

### 2.5. Data Analyses

According to the psychological factor influencing PHD or PDD during AMI and the delay-related processes, the included studies were grouped into 4 thematic macro-categories defined by the authors of the present systematic review as “Appraisal about AMI symptoms”, “Coping strategies”, “Alexithymia” and “Other psychological factors”.

## 3. Results

### 3.1. Search

The initial search identified 482 articles. Two hundred twenty-two duplicate articles were excluded. Abstracts from 260 studies were reviewed, and 194 were excluded for several reasons (128 were not pertinent; 12 were not written in English; 16 were reviews, 2 were meta-analyses, 28 were qualitative studies, 3 were editorials, 1 was a commentary, 1 was a case report and 3 resulted without an available full text). A total of 66 full articles were reviewed and 34 were excluded: 15 did not consider any psychological factors; five did not provide complete information about PHD or PDD delay; and 14 were not pertinent. Furthermore, four articles were added from the references. Finally, 36 studies were included in the qualitative synthesis (see Figure 1 and Appendix A).

### 3.2. Characteristics of the Included Studies

As shown in Appendix A, high heterogeneity relative to study design, samples size, psychological factors considered and measures to assess them did not permit comparison between findings with a meta-analysis. Among the included studies, 20 (55.6%) were cross-sectional, 7 (19.4%) were surveys, 8 (22.2%) were used as a case-control design and 1 (2.8%) was a cross-sectional with a case-control part. The samples size ranged from 47 to 913 subjects. Most of the studies refer to the sample as “AMI patients” without specifying if they included patients with STEMI only. Finally, most of the studies referred to PHD; only a few studies specifically examined the PDD.

Furthermore, the studies that focused their evaluation on PHD did not distinguish between PDD and the time needed for emergency medical system to treat the patient from the first intervention by a medicalized ambulance to hospital door arrival. However, we decided to also include these studies, trying to focus as much as possible to psychological factors and personality traits supposed to be linked to PDD.

### 3.3. Psychological Factors Impacting Pre-Hospital or Decisional Delay

The included studies were grouped into four macro-categories defined as “Appraisal of AMI symptoms”, “Coping strategies”, “Alexithymia” and “Other psychological factors”. These macro-categories employed these key psychological factors and processes related to PHD or PDD in order to map the main results of the included studies. Some studies, however, could fit in more than one category (see Appendix A).

#### 3.3.1. Appraisal of AMI Symptoms

This macro-category mainly refers to the mismatching between expected and experienced AMI symptoms and includes most of the studies (N 24).

A cross-sectional multi-center study with 799 STEMI patients [23] examined the association between expected and experienced AMI symptoms and its effects on patients’ care-seeking behaviors. Patients who interpreted their symptoms as having a non-cardiac origin were more likely to arrive at the hospital by self-transport and had longer PHD compared to those who did not.

A case-control study [24] reported that 315 patients with suspected AMI were more likely to arrive at the hospital within 4 h if they thought their symptoms might be a heart attack, if they believed that coronary heart diseases were preventable and if they took an ambulance to the hospital. The factor most strongly associated with early hospital arrival was the patient’s belief that the symptoms might represent a heart attack.

A multi-center descriptive survey involving 277 AMI patients [25] highlighted that cognitive and emotional responses to AMI symptoms affected patients’ decisions to seek treatment. Particularly, they did not appraise symptoms as serious or originating from their heart, had intermittent symptoms, waited to see whether symptoms disappeared, were worried about troubling others, feared what might happen if they sought treatment and did not realize the importance of symptoms. A survey conducted [26] among a sample of 317 AMI patients, found that PHD time increased by several cognitive and emotional processes such as the following: waiting to see if symptoms would go away, being too embarrassed to ask for assistance and not recognizing the gravity of symptoms. The independent predictors of increased PHD time included the following: not wanting to trouble anyone, the intermittent nature of symptoms and the failure to recognize them.

A cross-sectional study conducted with 145 AMI patients in Japan [27] revealed that wrong interpretations about AMI symptoms were significantly associated with PHD. Furthermore, a cultural factor linked to an individualism-collectivism dimension, defined as “independent and interdependent construal of self” [28], was associated with delay in accessing medical care. Differently from independent construal of self, the interdependent one is focused on social role, status and relationship to others such that individuals with ongoing AMI and with higher interdependent construal of self tend to make an effort to stay in their social role unless their symptoms interfere with the performance of the role and are faster in seeking medical help properly.

A cross-sectional study with 88 AMI patients [29] showed that most patients felt a mismatch between the experienced AMI symptoms and the expected ones. Patients tended to go to the hospital sooner if they experienced central chest pain and if their symptoms had rapid onset and matched prior expectations about what a heart attack would be like. Furthermore, these patients were more likely to call for help themselves rather than relying on others.

A multicenter descriptive survey [30] on 424 AMI patients pointed out that the patients who delayed longer showed the following cognitive and emotional responses: They appraised their symptoms as not serious, waited for symptoms to go away and worried about troubling others. Other factors associated with longer delay included the attribution of symptoms to a non-cardiac cause, inability to recognize symptoms as cardiac, embarrassment about seeking help and fear of the consequences of seeking help. Another multicenter cohort study with 595 AMI patients by the same research group [31] pointed out two independent predictors of PHD within an hour of symptoms’ onset: attribution of symptoms to the heart and not waiting for symptoms to go away.

A cross-sectional study [32] with 134 AMI patients found that, especially among men, predictors of delay included cognitive factors (i.e., waiting for symptoms to go away, no knowledge about AMI symptoms and no recognition of symptoms’ cardiac origin) along with emotional factors (i.e., anxiety due to symptom presentation, embarrassment in asking for help, worries about what may happen and about troubling others).

A cohort study [33] including 47 AMI patients noted that the discrepancy between the experienced and expected AMI symptoms was associated with a longer delay before reaching the hospital. Delay was reduced following conversation with someone about the symptoms during the symptom onset.

In a case-control study [34], 72 patients with AMI were divided into two groups: delayers (who waited more than 4 h before seeking help since experiencing their first chest pain) and prompt attenders. The patients who believed that they were having a heart attack sought help more quickly than those who did not. Furthermore, delayers had significantly lower scores on neuroticism and higher scores on denial and health locus of control (chance).

A cross-sectional survey [35] including 62 AMI patients pointed out that attribution of symptoms to heart disease—reflecting low denial and internal health locus of control—had a significant predictive effect on fast help seeking (within 60 min) while, interestingly, previous experience of heart disease did not.

A cross-sectional study [36], which analyzed the health beliefs and the delay in the care-seeking decisions of 79 patients with AMI, noted that 38% of patients delayed health care-seeking decisions (>1 h). Delayers perceived increased negative consequences of AMI and more barriers in the health care-seeking decision than non-delayers.

A cross-sectional study on 299 people with AMI [37] defined the “symptom incongruence” as the discrepancy between expected and experienced AMI symptoms before asking for medical treatment. It positively correlated with independently predicted PH delay. Furthermore, greater anxiety and greater perceived seriousness of symptoms were associated with less incongruence and shorter PHD.

A case-control study [38] among 453 people affected by AMI identified factors that distinguish “early responders” (those people who requested medical assistance within 60 min after the onset of AMI) from “late responders” (request made over 60 min after AMI symptom onset). PDD tended to decrease if patients believed that their symptoms were serious, cardiac in origin and they felt more anxious or upset when they started to notice symptoms, perceiving less control over them.

A cross-sectional survey with a case-control part [39] pointed out that, among the 306 patients with AMI who fulfill a validated questionnaire to measure patients’ appraisal, emotions and action tendencies preceding care seeking in AMI, the “perceived inability to act” (i.e., became paralyzed, lost control of themselves, became powerless, being frustrated and or unable to act despite persisting symptoms) and “symptom appraisal” regarding perceived symptom as not enough severe to seek medical care had the worst significant impact on PDD.

Another case-control study [19] involving 481 patients with AMI [17] pointed out that the factor increasing PDD was the consultation with a care provider different from emergency medical services at the first symptom presentation. Among these patients, 271 answered a questionnaire intended to explore the role of psychological factors different from behavioral and coping strategies adopted to manage the first symptoms. Analyses showed that “late callers” were those people who did not report sweat and those who felt that the situation was not severe.

Over one-third of the 720 patients with AMI involved in a cross-sectional study [40] arrived too late to the hospital, failing to recognize symptoms of AMI at their onset. Lack of knowledge of symptoms of heart attack and low severity of perceived chest pain were independently associated with PHD.

In a case-control study [41] among 165 AMI patients, the risk of delaying hospital presentation in patients who did not attribute their symptoms to a cardiac origin was approximately 6.9 times higher than in those who did. Patients with pre-infarction angina had approximately 3.6 times greater risk of delaying presentation (>6 h) than their counterparts. It was not specified whether this depended on memories of poor experiences with the emergency system or because of the belief that the event was not serious due to the experience of similar symptoms that were not diagnosed as AMI.

A cross-sectional study [42] underlined that 30% among 98 people with AMI experienced less severe symptoms than their expectations about the symptoms characterizing a heart attack. Some patients referred to the media’s tendency to portray AMI as a dramatic event. This lack of congruence resulted in a longer decision time for men but not for women.

A cross-sectional survey [43] involving 162 AMI patients highlighted that patients who did not perceive symptoms to be serious and failed to attribute them to a cardiac problem had significantly longer PHD.

A cross-sectional study [44] of 301 people with AMI pointed out that successful action relative to lower PHD relied on whether the patients who could correctly attribute the symptom experience to AMI were aware of their own susceptibility to the condition and had a good understanding of how the disease manifested itself. Positive cues from others in advising care seeking played favorable roles in promoting optimal care-seeking behaviors. The same authors [45] in a secondary data analysis involving the same sample (N = 301) pointed out that the perceived barriers to care seeking were the most significant predictor for longer PHD. Furthermore, poor congruence with what AMI symptoms were supposed to be and the occurrence of few typical AMI symptoms were also associated with longer PHD.

A cross-sectional study [46] involving 105 AMI patients showed that longer PHD was associated with the presumed ability to control symptoms and consequential dysfunctional coping strategies, such as taking medications, resting, trying to relax, praying or wishing symptoms would go away. Other factors associated with longer delay include misinterpretation of symptoms and lower perceived pain.

#### 3.3.2. Coping Strategies

This macro-category includes 14 studies focusing on the management of AMI symptoms and how medical help was sought. Among them, one study [47] also fitted the “Alexithymia” macro-category and is described in the related paragraph. Seven studies [19,25,26,39,45,46,48] also fitting the “AMI symptoms appraisal” macro-category have already been described in the previous section.

A descriptive study [49] conducted with 739 AMI patients pointed out that emotional attitudes to AMI symptoms and inadequate coping strategies are the major determinants of patients’ D delay. Ninety-three point three percent of the patients knew that AMI could be deadly. Among them, 43.9% decided to seek medical help too late (>1 h). The attitudes toward symptoms and coping strategies had the highest impact on late decisions.

Another descriptive study [50] on 135 AMI patients pointed out that the coping strategies most frequently used during ongoing AMI symptoms were as follows: trying to relax, wishing/praying for symptoms to disappear and discussing symptoms with someone. Coping strategies that showed a significant correlation with delay were as follows: trying not to think about symptoms; taking no-prescription medications; trying other home remedies; doing something to take one’s mind off the symptoms; engaging in normal activities; and convincing oneself and others that the problem was not serious.

A case-control study on 300 AMI patients [51] considered the association between coping strategies, defense mechanisms such as denial and the decisional delay just after AMI onset. The study showed that, during the acute phase of AMI, people who were quick in looking for medical care did not ignore their symptoms, sought less distraction and more social support, rightly interpreted the cardiac origin of the symptoms, had easing thoughts and denied to a lesser extent their feelings more frequently than delayers.

A recent descriptive study [48] on 135 AMI patients provided information about their history of AMI, congruence with symptom expectations, responses to symptoms, cardiac symptoms attribution and AMI care-seeking delay. The independent predictors of AMI care-seeking delay were wrong attributions of cardiac symptoms and emotion-focused coping. The total effect of symptom congruence on AMI care-seeking delay was significant.

An international cohort study with 913 AMI patients [52] showed that the median delay was too long to receive maximum the benefit from AMI therapies. Most participants wished or prayed symptoms would go away, tried to relax, pretended nothing was wrong, tried not to think about it, took medication, called a physician, tried self-help remedy, told someone nearby and reached the hospital by themself.

A single sample study with 61 AMI patients [53] noted that patients who perceived their AMI would have serious consequences had shorter delay than those who reported an emotional response with anxiety and surprise at symptoms onset. Patients with a strong active-cognitive coping style and patients with a strong problem-focused coping style had shorter delay times.

As for denial, a recent cross-sectional study on 533 AMI patients [54] showed that, in the time-window of 3–24 h regarding PH delay, denial accounted for a clinically relevant median excess time of 40 min in deniers compared to non-deniers.

#### 3.3.3. Alexithymia

This macro-category refers to the ability to recognize emotions and to communicate them properly to others and includes four studies. A study [34] involving 72 AMI patients (already cited in the “Symptoms appraisal” category) showed that there was no significant difference in alexithymia levels between “prompt attenders” and “delayers”, although they tended to be higher among “delayers”.

A second observational study [55] pointed out that reduced awareness in bodily sensation and emotions, typical of alexithymia, significantly predicted longer PH delay in 103 AMI patients. A “>4 h” cut-off was used for descriptive statistics, showing that 46% of 31 patients reached the hospital within 4 h. Longer interval time was not significantly associated with demographic or medical history categories or with Type A behavior.

In a case-control study [56] with 83 AMI patients, a “>120 min” was used as the cut-off to divide the sample into “Late Responders Case Group” (43.4%) and “Early Responders Control Group” (56.6%). Only higher alexithymia scores and contact with primary care were associated with increased PH delay. The co-presence of the two risk factors produced a cumulative risk without significant amplification.

A survey conducted among 95 people with AMI [47] underlined that the emotional response to the acute event and the ability to cope with stress are equally important factors in deciding to seek prompt medical help. High alexithymia was reported by 28% of the study population. Furthermore, high alexithymia, along with a non-productive coping strategy such as waiting for symptoms to go away, influenced patient responsiveness to cardiac symptoms, resulting in a significantly longer D delay.

#### 3.3.4. Other Psychological Factors

This macro-category includes two studies and refers to studies that pointed out the association between PDD or PHD and psychological factors such as the fear of death and the Type D personality trait.

A survey [57] focused on the association between fear of death (FoD) during AMI onset and PH delay. Among 592 AMI patients, 15% reported FoD with no difference between males and females. However, male patients reporting FoD had a 2.1-fold chance of early hospital arrival (<2 h) compared to their counterparts. This finding highlights that FoD does result in fearful immobilization but guides patients to optimal performance within the critical time window: they were more likely to perceive AMI symptoms as more severe, painful and life threatening than patients who did not report FoD.

A cross-sectional study [58] involving 256 patients with AMI pointed out that the Type D personality factor (characterized by negative affectivity and the tendency to experience negative emotions, as well as social inhibition) is associated with an increased awareness of the severity of AMI symptoms in female patients but not in male patients, resulting in less time spent on making decisions to seek care.

## 4. Discussion

Wrong appraisal and causal beliefs about symptoms, denial of symptoms and related emotions, external locus of control and dysfunctional coping strategies and behaviors were found to be related to longer PHD or PDD in people experiencing AMI.

These factors may suggest “externally oriented thinking”, which is relevant for alexithymia’s factor three and associated with PDD in AMI patients. On the other hand, fear of death, high personal control and Type D personality traits are associated with lower PH delay. It seems that the ability to recognize those emotions typically related to the onset of AMI is essential in order to perform the best behavioral strategies for managing such a critical situation. Alexithymia may be an overarching factor, which explains the disparate findings reported in the studies exploring the role of psychological factors in PDD in people with AMI.

### 4.1. Emotion Regulation in AMI

AMI is a potentially life-threatening stressful/traumatic situation. In this process, the time a patient takes to recognize an ongoing AMI, make a decision and seek help is crucial. As a life event, AMI requires affect regulation ability to recognize emotions, such as fear, to usefully cope with its symptoms and signs and to seek help properly. Prompt help-seeking places cardiologists in a position to treat AMI with the most effective available therapies, which require appropriate timing in order to ensure optimal outcomes.

Most of the reviewed studies did not directly examine emotional factors such as fear, anxiety or panic of death nor patients’ ability to recognize and mentalize them. These emotions are likely to influence the time taken during their decision-making process relative to seek medical help and causing delay. Poor consideration of the emotional component in the reaction to AMI symptoms could account for the failure of many prevention campaigns aimed at reducing PH or D delay [59]. Such failure may be imputed to a scarce focus on the different degrees of affect regulation ability in the target population: a prevention campaign might be unsuitable for people with high levels of alexithymia [18,34,47,55,56,60].

Dubayova et al. [6] confirmed the studies that directly related emotional factors—such as fear of death and high level of anxiety—to AMI onset [57,61]. They stated that fear is a factor that may push patients to look for help when experiencing the initial symptoms of an ongoing AMI. They examined the role of fear in determining earlier hospital presentation of patients with ongoing AMI and pointed out that delayers report lower levels of fear, a phenomenon that has been called “defensive bias”, “optimistic bias” or “denial”. This is not an illness-related phenomenon in itself but, when it appears as a pervasive and relatively rigid personality trait, it could also account for the ineffectiveness of certain (too frightening?) messages in prevention campaigns aimed at reducing the time to hospital presentation in the case of AMI [59].

Patients with greater levels of cardiac denial about the impact of ACS (Acute Coronary Syndromes) symptoms seem more likely to have longer PH delay due to the belief that their heart problem was caused by stress and emotional state and show an avoidant coping style [62]. Furthermore, Hwang et al. [63] identified the characteristics and interpretation of prodromal symptoms by patients with a first-time AMI and found that many did not recognize the importance of their warning symptoms. Therefore, only a few addressed a physician in response to any prodromal symptom.

Another descriptive study [64] pointed out the predictors of correct symptom attribution (CSA) during AMI. Investigation revealed that the patients whose AMI experience fitted their pre-existing symptom expectations were associated with greater odds of adopting a CSA. AMI patients with high level of alexithymia were likely to fail in identifying and communicating fear and other emotions linked to ongoing AMI and in distinguishing them from bodily sensations. In such a context, it could be “easier” not to recognize AMI, to deny AMI symptoms or underestimate them and, finally, to delay help-seeking. Regardless of delay time cut-off, a qualitative study [63] classified AMI patients’ profiles as “non-delayers”, “delayers” and “extended delayers” that seem to correspond to patients with “low alexithymia”, “intermediate alexithymia” and “high alexithymia”, respectively.

### 4.2. The Role Alexithymia in the Acute Phase of AMI

It was pointed out that general anxiety disorder (GAD) is highly associated with acute anxiety and fear of death as well as lower PH delay. Patients with GAD are more likely to perceive a higher cardiovascular risk, which results in a higher likelihood of autonomously deciding to arrive at the hospital during the acute phase of AMI. However, GAD was also highly associated with impaired psychological well-being, stress and fatigue [61]. A form of “secondary alexithymia” [65] could be part of a “spectrum of post-traumatic disturbances” and could contribute to PH delay during AMI or ACS, especially among individuals with a history of AMI and/or coronary heart disease (CHD) [66,67]. These patients need support in understanding the right skills to cope with the emotional arousal surrounding the primary, as well as secondary, traumatic experience of symptoms and signs and the tertiary trauma from the long-term consequences of CHD [66,67]. When assessed, higher levels of alexithymia are associated with higher delay time in seeking medical help for AMI [18,34,47,55,56,60]. In the study of O’Carroll et al. [34], the findings concerning the difference in alexithymia levels between “prompt attenders” and “delayers”, although not reaching statistical significance, are interesting. Alexithymia levels are higher in “delayers” than in “prompt attenders”, but the threshold selected to identify a “prompt attender” was too long (<4 h). Alexithymia may influence illness behavior, particularly the experience and reporting of physical symptoms and treatment seeking [11,12,13]. AMI is often unrecognized either by patients with high alexithymia who fail to notice or properly report the event to their physician or by the physician who could fail to diagnose it. Theisen et al. [60] found that patients with unrecognized AMI showed greater alexithymia as well as a greater belief that their health was determined by chance-related factors. Evidence from a brain-imaging study performed using PET and MRI [68] indicates that, during the evaluation of response to emotional stimuli, subjects with a high level of alexithymia have reduced flow and brain activity in some regions of the cingulate cortex. Reduced grey matter in the cingulate cortex has been reported in patients with coronary artery disease (a risk factor for AMI), which points to a link between defective cognition on emotion regulation and AMI [69].

## 5. Conclusions

The association between PH or D delay and psychological factors as well as personality traits, particularly alexithymia, has not been reliably and systematically assessed relative to care-seeking behavior in patients with ongoing AMI. Meta-analysis of the data was prevented by the heterogeneity of methods and measures. It could be hypothesized that patients who are more capable of identifying inner experiences of emotions and bodily sensations are more likely to seek earlier and proper treatments for AMI. Some interventions to reduce alexithymia have been proven effective in patients with coronary disease [70]. Such interventions may also be offered to patients with risk factors for AMI in order to explore their potential preventive utility relative to pre-hospital and patients’ decisional delay. Further research is needed to fully establish the strength of the influence of alexithymia on delay by a parallel analysis of other factors, such as the degree of depression and general intellectual decline as a result of organic changes in the central nervous system.

## Figures and Tables

**Figure 1 jcm-10-03813-f001:**
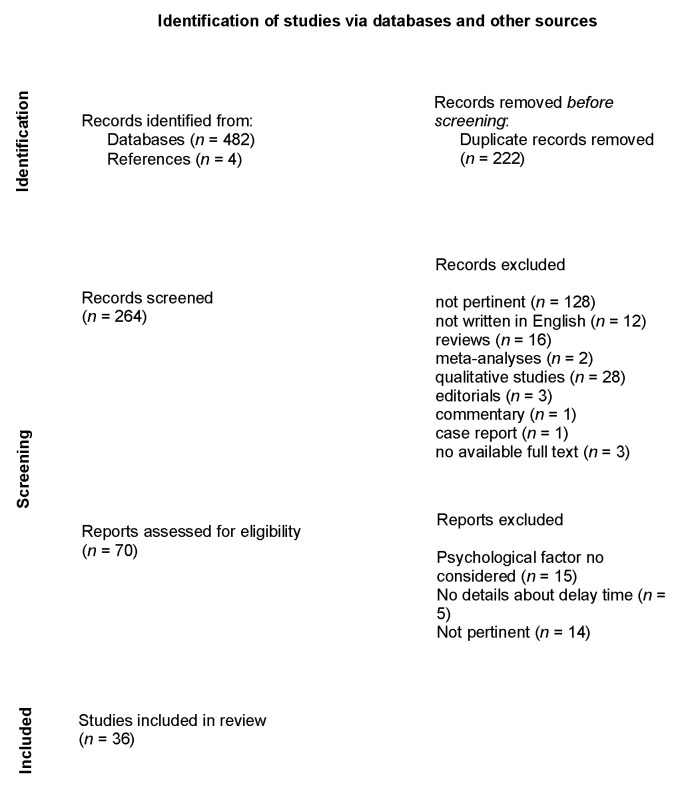
PRISMA diagram—systematic search and study selection process [22].

## Data Availability

Not pertinent for a review.

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
