# Peer review of "Surviving to Acute Myocardial Infarction: The Role of Psychological Factors and Alexithymia in Delayed Time to Searching Care: A Systematic Review"

_jcm, 2021, doi:10.3390/jcm10173813_

Round 1
Reviewer 1 Report
The analyzed publication is an overview analysis of a wide range of publications. The conclusions are of practical importance and encourage more extensive research on the psychological factors determining the delay in starting AMI treatment. An interesting conclusion of the study is the indication of alexithymia as one of the causative factors. The strength of the influence of this factor, however, can only be fully determined after a parallel analysis of other factors, such as the degree of depression and general intellectual decline as a result of organic changes in the CNS.
Author Response
Response to Reviewer 1 Comments
Point 1: Comments and Suggestions for Authors
The analyzed publication is an overview analysis of a wide range of publications. The conclusions are of practical importance and encourage more extensive research on the psychological factors determining the delay in starting AMI treatment. An interesting conclusion of the study is the indication of alexithymia as one of the causative factors. The strength of the influence of this factor, however, can only be fully determined after a parallel analysis of other factors, such as the degree of depression and general intellectual decline as a result of organic changes in the CNS.
Response 1: we add the following statement in the Conclusions section page 12 lines 587-590: “Further research is needed to fully establish the strength of the influence of alexithymia on delay by a parallel analysis of other factors, such as the degree of depression and general intellectual decline as a result of organic changes in the central nervous system.”

Reviewer 2 Report
This is a very interesting paper which highlights an important psychological dimension to the issue of delayed time in seeking help for AMI. It fills a gap in the literature very nicely. There are a few small grammatical matters to address which I note below, but the only more substantive comment that I have is that I understand why you selected studies with control groups but I think a bit more detail in your discussion of selection criteria to justify exclusion of qualitative studies is warranted.
Small items to fix:
The funding section is a repeat of the author contribution section.
It would be useful to add a reference or 2 to support the statement at line 48 regarding mortality and morbidity.
line 35 ... culprit coronary artery...
line 120 ...study, and consistent with...
line 133 In case control studies, also reported were cut-off scores to define...
line 135 ...factors presumed to be associated with PHD...
line 256 ...symptoms before asking for medical...
line 260 ...453 people affected by AMI...
line 267 ...fulfill a validated questionnaire...
line 275 ...at the first symptom presentation.
line 276 ...271 answered a questionnaire
line 298 ...study [41] of 301 people...
line 303 same authors [42] ...the same sample [N=]
line 315 ...and how medical help was sought...
line 316 ...described in the previous section.
line 356 ...ability to recognise emotions...
line 374 ..along with a non-productive coping strategy...
line 379 ...such as the fear of death
line 401 ...to manage such a critical situation.
line 410 ...puts cardiologists in a position to treat AMI...
line 417 ...to a scarce focusing on the different...
Author Response
Response to Reviewer 2 Comments
This is a very interesting paper which highlights an important psychological dimension to the issue of delayed time in seeking help for AMI. It fills a gap in the literature very nicely. There are a few small grammatical matters to address which I note below, but the only more substantive comment that I
Point 1: The funding section is a repeat of the author contribution section.
Response 1: We agree with the Referee indication and we have removed the funding section page 13
Point 2: It would be useful to add a reference or 2 to support the statement at line 48 regarding mortality and morbidity.
Response 2 As suggested, we added the following references:
[3] De Luca G, Suryapranata H, Ottervanger JP, Antman EM. Time delay to treatment and mortality in primary angioplasty for acute myocardial infarction: every minute of delay counts. Circulation 2004; 109: 1223-5.
[4] Wijns W, Naber CK. Reperfusion delay in patients with high-risk ST-segment elevation myocardial infarction: every minute counts, much more than suspected. Eur Heart J 2018; 39: 1075-7
Furthermore, we numbered again the other References in the text, as well as in the Table 1.
Response 3: As suggested we changed:
Page 1 line 35 ... culprit coronary artery...
Page 3 line 129 ...study, and consistent with...
Page 4 line 150 In case control studies, also reported were cut-off scores to define...
Page 4 line 152 ...factors presumed to be associated with PHD...
Page 7 line 284 ...symptoms before asking for medical...
Page 7 line 288 ...453 people affected by AMI...
Page 7 line 297 ...fulfill a validated questionnaire...
Page 8 line 315 ...at the first symptom presentation.
Page 8 line 316 ...271 answered a questionnaire
Page 8 line 338 ...study [41] of 301 people...
Page line 341-342 same authors [42] ...the same sample [N=]
Page 9 line 367 ...and how medical help was sought...
Page 9 line 370 ...described in the previous section.
Page 10 line 425 ...ability to recognise emotions...
Page 10 line 443 ..along with a non-productive coping strategy...
Page line 448 ...such as the fear of death
Page 11 line 480 ...to manage such a critical situation.
Page 11 line 490 ...puts cardiologists in a position to treat AMI...
Page 11 line 497 ...to a scarce focusing on the different...
